# Exploration of a novel geoengineering solution: lighting up tropical forests at night

Xueyuan Gao[1], Shunlin Liang[1]*, Dongdong Wang[1], Yan Li[2], Bin He[3], Aolin Jia[1]

[1] Department of Geographical Sciences, University of Maryland, U.S.

[2] Faculty of Geographical Science, Beijing Normal University, China

[3] College of Global Change and Earth System Science, Beijing Normal University, China

*Corresponding author. Email: sliang@umd.edu

**Abstract**

Plants primarily conduct photosynthesis in the daytime, offering an opportunity to increase photosynthesis and carbon sink by providing light at night. We used a fully coupled Earth System Model to quantify the carbon sequestration and climate effects of a novel carbon removal proposal: lighting up tropical forests at night via lamp networks above the forest canopy. Simulation results show that additional light increased tropical forest carbon sink by $10.4 \pm 0.05$ petagrams of carbon per year during a 16-year lighting experiment, resulting in a decrease in atmospheric $CO_2$ and suppression of global warming. In addition, local temperature and precipitation increased. The energy requirement for capturing one ton of carbon is lower than that of Direct Air Carbon Capture. When the lighting experiment was terminated, tropical forests started to release carbon slowly. This study suggests that lighting up tropical forests at night could be an emergency solution to climate change, and carbon removal actions focused on enhancing ecosystem productivity by altering environmental factors in the short term could induce post-action $CO_2$ outgassing.

**Short summary**

Numerical experiments with a coupled Earth System Model show that large-scale nighttime artificial lighting in tropical forests will significantly increase carbon sink, local temperature, and precipitation, and requires less energy than Direct Air Carbon Capture for capturing 1 ton carbon, suggesting that it could be a powerful climate mitigation option. Side effects include the $CO_2$ outgassing after the termination of the nighttime lighting and the impacts on local wildlife.

**Keywords**: climate change; Earth system model; geoengineering; carbon cycle; tropical forests

## 1. Introduction


Anthropogenic greenhouse gas (GHG) emissions have led the global mean temperature to
increase by approximately 1.1 degree Celsius since the industrial revolution(IPCC, 2013, 2018;
IPCC AR6 WGI, 2021). Changes in climate have caused impacts on natural ecosystems and
human societies, such as mass ice sheet melt(Jevrejeva et al., 2016), devastating heat
waves(Dosio et al., 2018), and increase in extreme climate events(Kirchmeier-Young and Zhang,
2020), exposing natural and human systems to uncertainties and the risks of unsustainable
development(Gao et al., 2019, 2020). Despite the scientific consensus on climate change,
emission-reduction efforts have made slow or little progress with global GHG emissions
continuing to rise(IPCC AR6 WGI, 2021). In this context, geoengineering options are
increasingly being considered as means of deliberately intervening in Earth's climate system in
the second half of the 21st century(IPCC AR6 WGI, 2021; Moore et al., 2015).
Existing geoengineering proposals tend to align with two fundamentally different strategies:
Solar Geoengineering (SG)(Abatayo et al., 2020; Proctor et al., 2018; Robock et al., 2009) and
Carbon Capture and Sequestration (CCS)(IPCC, 2005; Jones, 2008; Leung et al., 2014). SG and
related techniques reduce the amount of incoming radiation from the sun typically via
stratospheric aerosol injection, subsequently affecting the planet's temperature. Although they
may be able to offset temperature increase rapidly, previous studies indicate the potential for
political instability(Abatayo et al., 2020) and negative impacts on human health(Robock et al.,
2009) and agriculture(Proctor et al., 2018). Comparatively, CCS removes carbon from the global
carbon cycle by artificial machines and saves it for long-term storage or for industrial
reutilization(IPCC, 2005). While technically feasible, the environmental risks for the transport
and storage of $CO_2$, limited carbon storage capability, and high cost remain large obstacles of
implementing CCS(IPCC, 2005; Jones, 2008; Leung et al., 2014).
In this study, the authors propose a novel geoengineering solution: lighting up tropical forests at
night by installing lamp networks above the forest canopy(Graham et al., 2003), which lengthens
photoperiods and leads to greater photosynthesis and carbon sequestration, and helps mitigate
climate change. Contrasting to traditional CCS techniques, this strategy utilizes nature carbon
sink to capture and sequester $CO_2$ from air and avoids long-distance transport and geological
storage.
Structurally intact tropical forests are by far the most efficient carbon-capture method(Mitchard,
2018), and they act as an important carbon sink against rising $CO_2$ levels(Pan et al., 2011;
Sullivan et al., 2020). Although intact tropical forest growth is likely suffering from warming
and moisture stress induced by anthropogenic greenhouse gas emissions(Aguirre-Gutiérrez et al.,
2020; Doughty et al., 2015; Gatti et al., 2021; Hubau et al., 2020), light is still the primary factor
limiting tropical tree growth due to cloud cover, especially during the rainy season(Boisvenue
and Running, 2006; Graham et al., 2003). Studies on the photoperiodic control of tropical trees'
growth typically fall into two categories: physiological field observations under seasonal
variations of day length(Borchert et al., 2005; Pires et al., 2018; Rivera et al., 2002), and
physiological greenhouse observations under experimental variations of the photoperiod(Dixit
and Singh, 2014; Djerrab et al., 2021; Luo et al., 2021; Stubblebine et al., 1978). Field
observations have shown that longer photoperiods facilitate the bud break and flowering in
tropical forests(Borchert et al., 2005; Pires et al., 2018; Rivera et al., 2002). Greenhouse
experiments either lengthen or shorten photoperiods, and results suggest that short photoperiods
reduce plant growth rate and lead to thinner leaves and lower chlorophyll content(Djerrab et al.,
2021; Luo et al., 2021), while long photoperiods increase stem growth rate and stimulate tree
growth(Dixit and Singh, 2014; Stubblebine et al., 1978). These studies are more focused on
specific tropical plant species and tend to agree that longer photoperiods might have a positive
effect on vegetative growth in tropical forests.  Ecosystem-level field experiments are critical for
taking into account key environmental factors that are missing in greenhouse experiments (e.g.
water and nutrition constraints), and for informing model parameterizations, although they are
lacking so far.
Earth System Models provides state-of-the-art computer simulations of key processes and
climate states across the Earth(Danabasoglu et al., 2020). In this study the authors used a fully
coupled Earth System Model, Community Earth System Model version 2 (CESM2) developed
by the U.S. National Center for Atmospheric Research(Danabasoglu et al., 2020), to test the
carbon sequestration and climate effects of this geoengineering measure by conducting
numerical lighting experiments. Briefly, we added additional diffuse visible light to tropical
forest canopy at night (see Supplementary Figure 1) assuming that trees will receive light from
multiple directions (e.g., multiple lamps). Tropical forest grids were defined by "Broadleaf
Evergreen Tree Area Percentage" being greater than 60% between 20°N and 20°S. The lighting
experiment started from 12:00 am on January 1st, 2015 (UTC time), and the simulation exercise
was conducted across numerous timescales and lighting levels:
(1) Historical control simulation from 2001 to 2014
(2) 24-hour lighting experiment with various lighting powers on January 1st, 2015
(3) 16-year lighting experiment with the optimal lighting power from 2015 to 2030
(4) 20-year simulation after the experiment termination from 2031 to 2050
(5) Future control simulation from 2015 to 2050
Both experiment and control simulations in the future from 2015 to 2050 were on top of the
Shared Socioeconomic Pathways (SSP) 126 scenario(Riahi et al., 2017). Each simulation has a
spatial resolution of 1° and has two members (created from small perturbations to initial
conditions) to provide uncertainty estimation. (see Methods for detailed experimental design)

## 2. Results

2.1 24-hour lighting experiment with various lighting powers on January 1[st] 2015

Figure 1 shows the changes in carbon and energy fluxes of Amazonian tropical forests for 24 hours since the start of the nighttime lighting experiment at 12:00 am January 1[st], 2015 (UTC time; See Supplementary Figure 2 and 3 for African and Asian tropical forest responses). Tropical forests had a significant response to nighttime radiation, but the response was different under 100, 200, 300, and 400 $W/m^2$ lighting powers. The lighting experiment altered the nighttime energy balance and increased near surface air temperature, latent heat, and sensible heat. Higher lighting powers led to greater increases in air temperature, latent heat and sensible heat. Meanwhile, the additional light activated photosynthesis and increased Net Ecosystem Productivity (NEP). Nighttime NEP reached the peak at $200W/m^2$ and seemed to be suppressed when the lighting power was higher. Comparison of NEP across lighting powers suggests that $200W/m^2$ is optimal in terms of activating additional photosynthesis. The nighttime NEP is higher than daytime because nighttime surface radiation is solely diffuse visible light while daytime surface radiation is composed of direct NIR (~16%), diffuse NIR (~30%), direct visible light (~15%), and diffuse visible light (~39%). African and Asian tropical forests showed similar responses.

During daytime in the control simulation, the maximum NEP shows up around 9:00-11:00 am (Fig. 1-b). It is not likely to be due to clouds according to the diurnal pattern of the surface downward shortwave radiation (Fig.1-a). We examined the diurnal curve of the soil moisture (the red dash line in Fig. 1-b), and it seems to be due to soil moisture stress. Soil moisture was consumed quickly in the morning which led to water stress for plant growth in the afternoon. The soil moisture pattern also explains the biased distribution of daytime surface air temperature (Fig.1-c), and slightly biased daytime latent heat (Fig.1-d), and daytime sensible heat (Fig.1-e).

**Fig. 1.** Amazonian tropical forest responses for 24 hours since the start of the nighttime lighting
experiment at 12:00 am January 1st, 2015 (UTC time) under various nighttime lighting powers.
Panel (a) refers to surface downward shortwave radiation. Nighttime NEP (b) reached the peak at
200W/m$^2$, suggesting that 200W/m$^2$ is optimal in terms of activating additional photosynthesis.

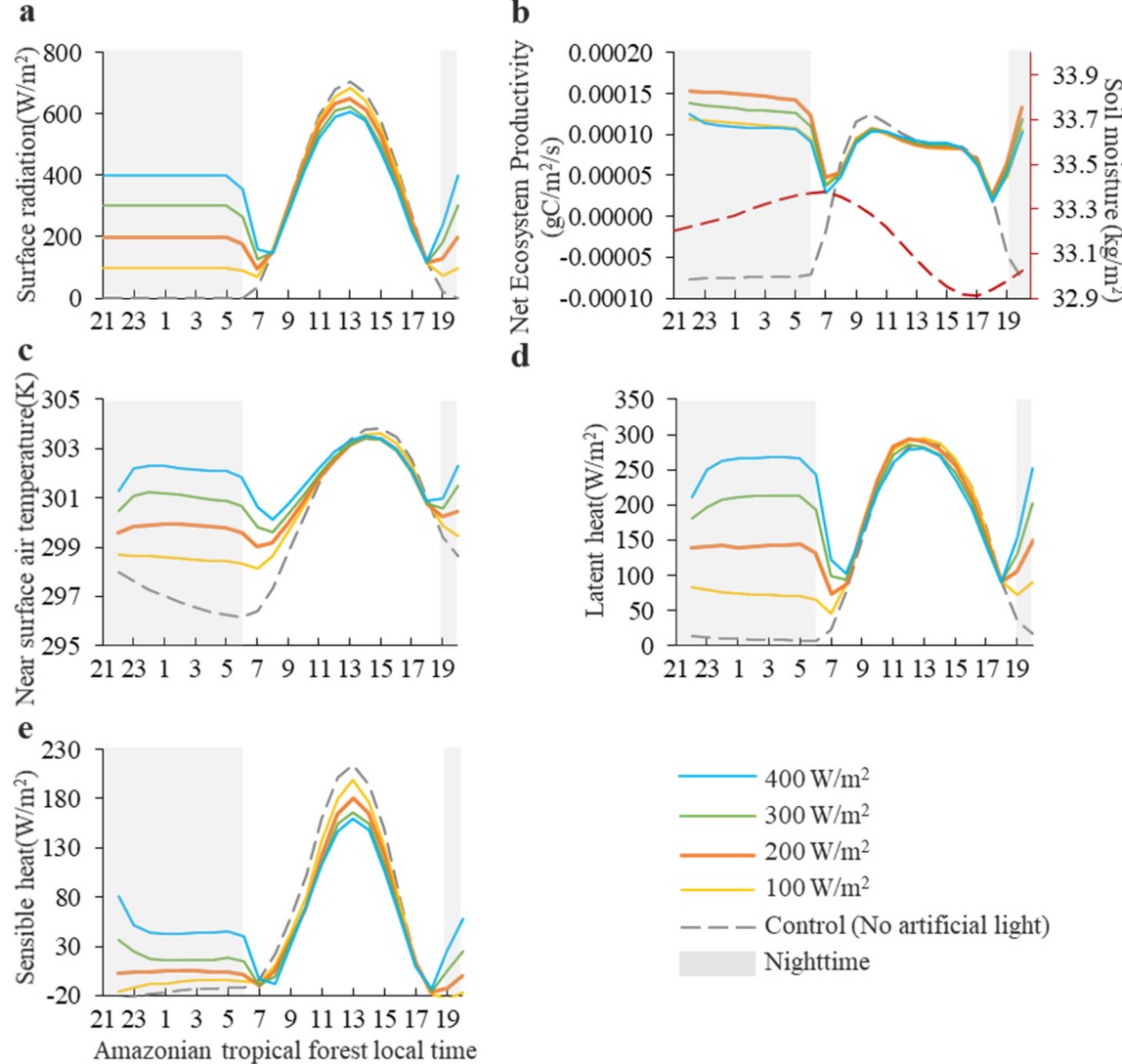


2.2 16-year lighting experiment with the optimal lighting power from 2015 to 2030
The yellow lines in Figure 2 show that tropical forest carbon fluxes and climates were
significantly altered by a 16-year continuous lighting experiment at night with a 200W/m$^2$
power. The annual gross primary production and autotrophic respiration increased by twice near
instantaneously, while the heterotrophic respiration had a slower response and increased
continuously over a longer period. We purport these changes to be due to the increase in local
temperature and the gradual accumulation of organic matter in the soil. Simulation results show
that the lighting experiment also decreased wildfire emissions as soil moisture increased despite
the expansion of the coarse woody debris and litter carbon pool that provides potential burning
materials. Overall, the net carbon uptake increased to around 25 petagrams of carbon per year
(Pg C yr$^{-1}$) in the beginning of the lighting experiment, although it decreased with time due to
the continuous increase in heterotrophic respiration. The lighting experiment increased the net
carbon uptake in tropical forests by 15.3 times over the simulation period (from 0.68$\pm$0.02 Pg C
yr$^{-1}$ over 2001-2014 to 11.1$\pm$0.05 Pg C yr$^{-1}$ over 2015-2030). Among all the absorbed carbon,
75% entered the vegetation carbon pool, 16% entered the coarse woody debris and litter carbon
pool, and 9% entered the soil carbon pool (Figure 3-b).
Simulation results show that local climates were also significantly impacted (Figure 2-g,h). The
annual average air temperature increased by around 1.3℃, and annual precipitation almost
doubled. The temperature and precipitation increase showed no significant seasonal trend
(Supplementary Figure 4). Globally, the atmospheric $CO_2$ concentration dropped quickly in the
first several years, while turned flat in the latter of the lighting experiment. As a result, the global
average air temperature increase was suppressed by around 0.5℃.
Amazonian, African, and Asian tropical forests present different capabilities to offset annual
atmospheric carbon accumulation during the lighting experiment (Figure 4). In the current global
carbon budget(Friedlingstein et al., 2019) (averaged from 2009 to 2018), approximately 11$\pm$0.5
Pg C yr$^{-1}$ was released into atmosphere by anthropogenic activities including fossil fuel
combustion and land use, among which 2.5$\pm$0.6 Pg C yr$^{-1}$ was absorbed by ocean, 3.2$\pm$0.6 Pg C
yr$^{-1}$ was absorbed by land, and 4.9$\pm$0.02 Pg C yr$^{-1}$ was accumulated in atmosphere resulting in
the concerned warming and climate change. The lighting experiment enhanced Amazonian
tropical forest net carbon uptake to 6.5$\pm$0.04 Pg C yr$^{-1}$ (averaged during 2015 to 2030),
suggesting that lighting up Amazonian tropical forests along could completely offset
anthropogenic carbon emissions. African and Asian tropical forests showed lower capabilities
with the net carbon uptake being approximately $2.0\pm0.002$ and $2.6\pm0.008$ Pg C yr$^{-1}$ respectively
(see Supplementary Figure 5, 6, and 7 for Amazonian, African, and Asian tropical forest carbon
flux, carbon amount, and climate responses respectively).


















**Fig. 2.** Global tropical forest carbon flux and climate responses under and after the lighting
experiment. Ta in panel (g & k): Near surface air temperature. Soil moisture in Panel (i) refers to
the mass of water in the 10cm soil surface. Shaded areas represent uncertainties.

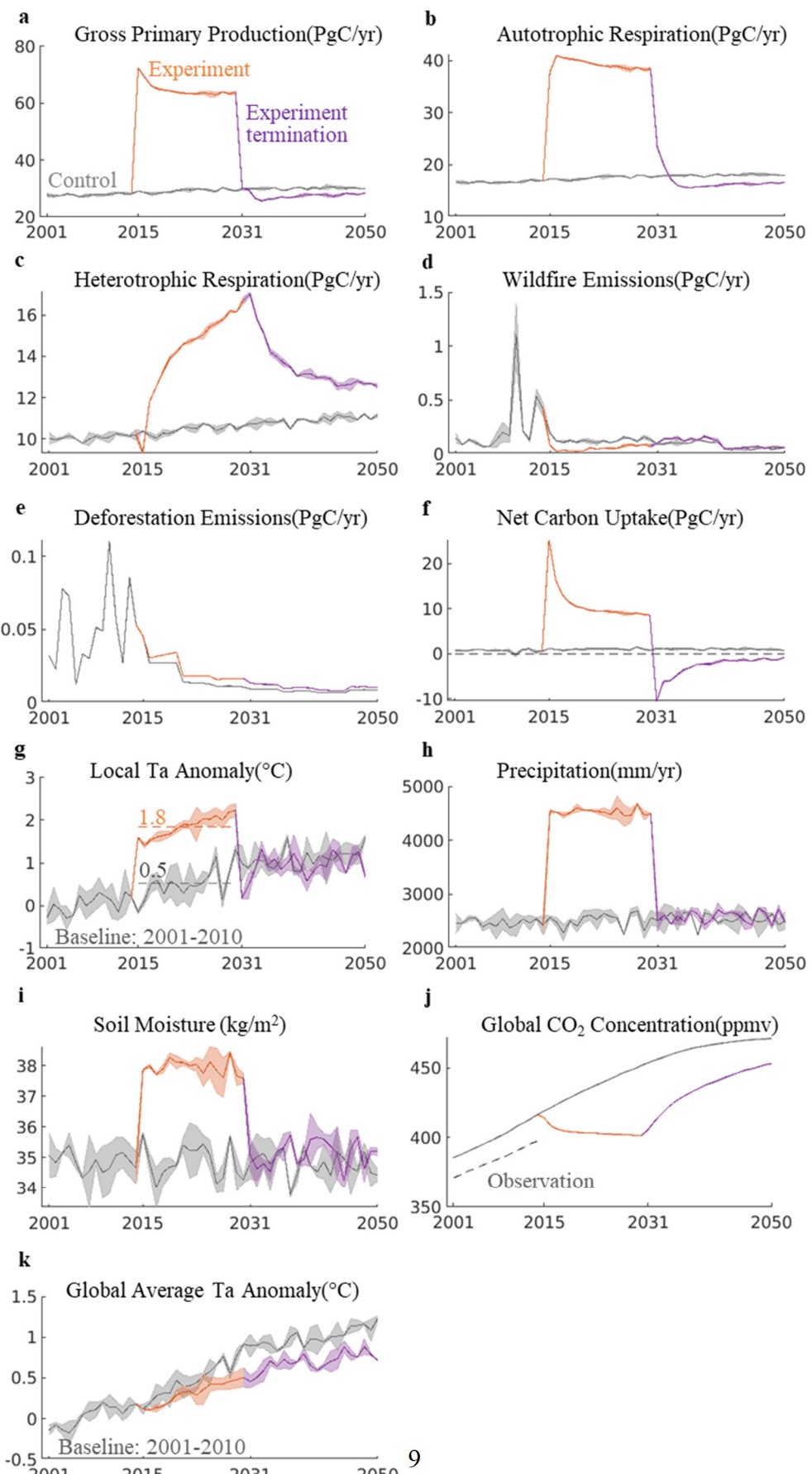

189                                                       9

**Fig. 3.** Where Did the Net Absorbed Carbon Go? Global Tropical Forest Carbon Amount Responses. Panel (a): the current carbon amount in different carbon pools. Panel (b): carbon amount in 2030 after 16-year lighting experiments. Panel (c): carbon amount in 2050 after 20 years since the termination of the lighting experiments. The solid circles in panel (b) and (c) refer to carbon amount changes with respect to panel (a). The numbers in panels (a-c) are based on panels (d-f). Tree drawing courtesy of © Ning Zeng.

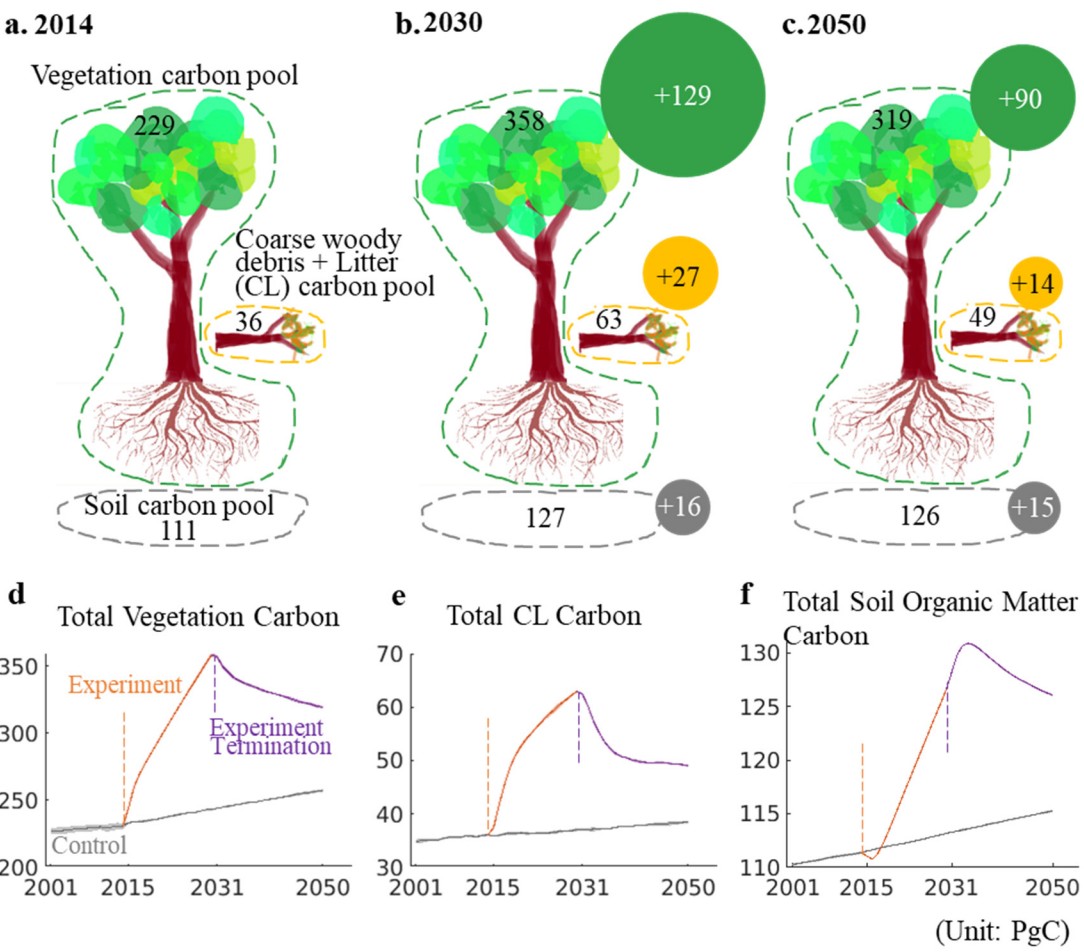

**Fig. 4.** Capabilities of Amazonian, African, and Asian tropical forests to offset annual
atmospheric carbon accumulation.

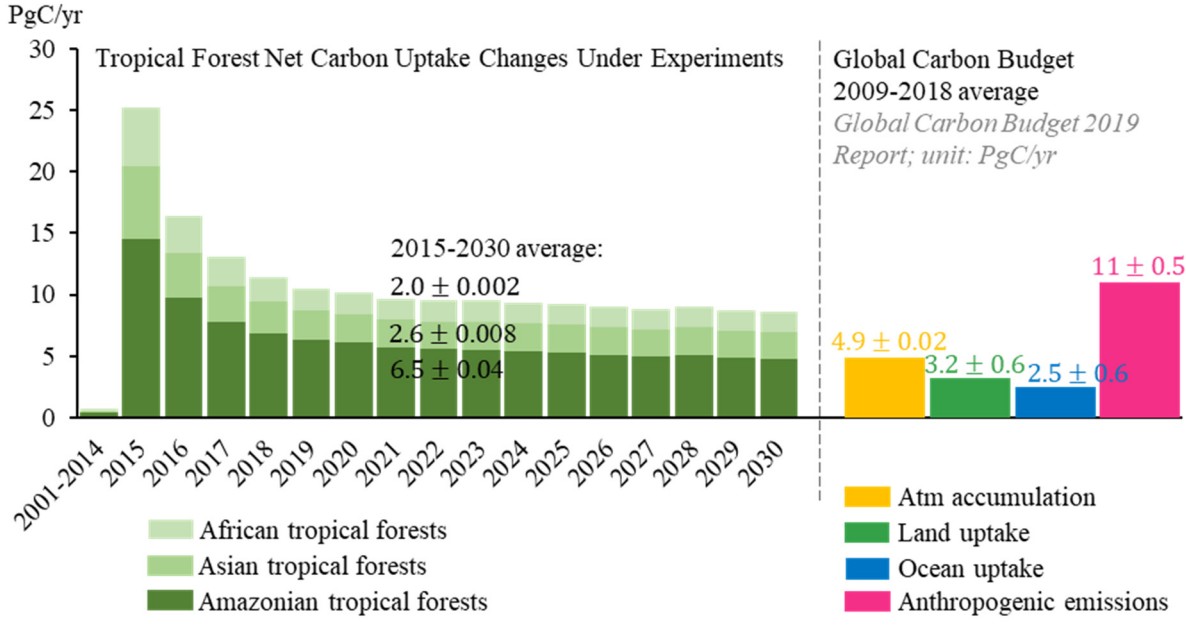



We estimated the energy requirement of this strategy for capturing one ton of carbon (see
Methods), and compared it to that of Direct Air Carbon Capture (DACC) estimated by recent
studies(Chatterjee and Huang, 2020; Realmonte et al., 2019). As the carbon uptake efficiency of
the tropical forest ecosystem decreases with time when under consecutive nighttime lighting, the
energy requirement for capturing one ton of carbon increases (Figure 5 purple line).
Nevertheless, the energy requirement of this strategy is lower than that of DACC, or is
equivalent to the most optimistic estimation of DACC's energy requirement that excludes the
energy costs required for carbon transport, storage, and utilization. (see Discussion)

**Figure 5.** Energy requirement for DACC and the nighttime lighting strategy

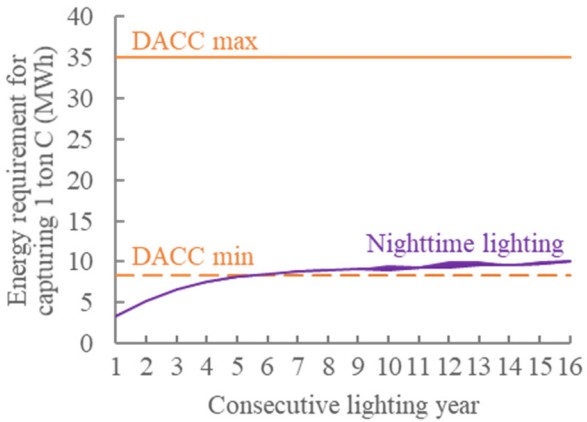


2.3 20-year simulation after the experiment termination from 2031 to 2050
The lighting experiment was terminated at 12:00 am January 1st, 2031 (UTC time), and model
simulations continued for 20 years to 2050 (see the purple lines in Figure 2). The annual gross
primary production and autotrophic respiration dropped quickly, ultimately reaching levels that
were even lower than the control period due to a reduction in atmospheric $CO_2$ ($CO_2$ has
fertilization effect in the model). Heterotrophic respiration remained high and decreased much
slower at a speed 10 times lower than gross primary production and autotrophic respiration. The
soil organic matter carbon pool continued to expand due to the entering of litter carbon during
the first 2-3 years following the experiment termination (Figure 3-f). The vegetation carbon pool
shrunk as trees produced less leaves (Figure 3-d). As a result, tropical forests turned into a net
carbon source and remained so until the end of the simulation in 2050 (Figure 2-f). 31.4% of the
carbon that had been absorbed during the lighting experiment was released back to the
atmosphere. This number would likely be higher if the simulation continued. As a result, the
global atmospheric $CO_2$ concentration returned to a level slightly lower than the control scenario.
Local air temperature and precipitation returned to control levels.

## 3. Discussion

Physiological responses of tropical trees to longer photoperiods at the ecosystem level remains one of the biggest uncertainties in model simulations. Some field experiments indicate that higher $CO_2$ did not increase carbon sequestration of forests without added nutrients(Oren et al., 2001), suggesting tree growth might be limited by nutrient supply. The simulated local warming might also suppress tree growth(Gatti et al., 2021). Some observational evidence shows that intact tropical forest carbon sinks have been negatively influenced by warming and moisture stress(Doughty et al., 2015; Gatti et al., 2021) and might be reaching saturation(Hubau et al., 2020). However, the model predicted increase in precipitation and soil moisture, and previous studies have shown hydro climate plays a key role in deciding the effects of warming on tree growth(Guan et al., 2015; Reich et al., 2018). No direct evidence exists to verify the simulation results. Ecosystem-level field experiments are needed to understand how tropical forest ecosystems respond to longer photoperiods.

CESM2 likely overestimated the local air temperature increase in tropical forests for the omission of chemical energy stored during photosynthesis(Sellers, 1992). In CESM2 and other modern Earth System Models(Sellers, 1992), the canopy energy equation(Danabasoglu et al., 2020) uses the solar radiation absorbed by the vegetation to calculate temperature:

$$-\vec{S}_v + \vec{L}_v(T_v) + H_v(T_v) + \lambda E_v(T_v) = 0 \qquad (1)$$

where $\vec{S}_v$ is the solar radiation absorbed by the vegetation, $\vec{L}_v$ is the net longwave radiation absorbed by vegetation, and $H_v$ and $\lambda E_v$ are the sensible and latent heat fluxes from vegetation, respectively. $\vec{L}_v, H_v$, and $\lambda E_v$ depend on the vegetation temperature $T_v$.

The chemical energy that is stored during photosynthesis and released by respiration is ignored as the net chemical energy usually amounts to less than 1% of absorbed insolation (around 0.6%(Trenberth et al., 2009)). In our lighting experiment from 2015 to 2030, however, 17% of absorbed insolation was fixed in the ecosystem as chemical energy (Figure 2-f) and did not contribute to local air temperature increase. The model failed to exclude this chemical energy storage from the energy equation. Therefore, the model overestimated the local temperature increase. This suggests that the temperature simulation results should be treated carefully when Earth System Models are used to do extreme scenario experiments associated with biogeochemistry.

Tropical forests experienced significant increase in carbon sink during the lighting experiment,
but ultimately transitioned from a sink to a source after the experiment was terminated (Figure 2-
f). Studies(Koven et al., 2021; Tokarska and Zickfeld, 2015) investigating the effects of
overshoot future scenarios (positive carbon emissions followed by net-negative emissions) on
terrestrial carbon cycle have observed similar phenomenon. During a positive emissions phase,
terrestrial carbon cycles tend to absorb some fraction of added $CO_2$; however, during a removal
phase they tend to release $CO_2$. The mechanism of these phenomena is the different responding
rates of vegetative primary productivity and heterotrophic respiration to lengthening and
shortening photoperiods, or increasing and decreasing atmospheric $CO_2$, with primary
productivity responding much quicker than heterotrophic respiration. It is understandable when
considering the diurnal pattern of forest carbon uptake. In the daytime, forests act as a carbon
sink because photosynthesis is greater than respiration. In the nighttime respiration continues
while photosynthesis abates, making forests a carbon source. Additional light/$CO_2$ would
increase carbon sink by increasing both photosynthesis and respiration (sometimes referred to as
a fertilization effect). When the additional light/$CO_2$ is removed, photosynthesis decreases
quickly while respiration remains high, making forests a greater carbon source. It suggests that
carbon removal actions focused on enhancing ecosystem productivity by altering environmental
factors in the short term could induce this post-action $CO_2$ outgassing.
The global average surface air temperature remained below the control level after the termination
of lighting experiments due to a reduction in atmospheric $CO_2$ concentration (Fig.2-k). However,
the local air temperature went back to the control level and seems to be not influenced by $CO_2$
reduction (Fig.2-g). We attribute it to two possible reasons. First, different regions tend to have
diverse air temperature responses to global $CO_2$ changes. Arctic regions show a much larger
temperature increase in response to $CO_2$ increase, while the temperature increase in tropical
regions is not that significant. Similarly, the $CO_2$ reduction may exert diverse impacts on
temperature changes in different regions. Second, the temperature change in tropical forests at
the termination of the experiment is controlled by two factors in this study, decreased incoming
shortwave radiation and reduced $CO_2$. The former has a much larger impact on the local energy
balance than the latter. Therefore, the influence of $CO_2$ reduction on local tropical forests is not
as large as on the global scale.
Large clean energy requirements have always been a hurdle to large-scale deployment of any
Carbon Dioxide Removal (CDR) techniques, including DACC and the strategy we discuss in this
study. Our estimation suggests that the energy requirement of this strategy for capturing one ton
of carbon is less than that of DACC. Specifically, if we give DACC 100 units of energy
(100MWh) per year, DACC could fix 3-12 ton carbon per year. If we give forests 100 units of
extra energy per year, forests could fix around 19.5 ton carbon per year on average (15-year
average: 29 ton carbon in the first year and 10 ton carbon in the 15th year due to an increase of
soil respiration); however, only 17 units of energy are actively used to fix carbon, and the rest 83
units of energy end up as heat which increases local temperature. Therefore, the energy use
efficiency of this strategy is low, which is a major drawback.
Other than the direct lighting energy, this strategy requires additional energy associated with
manufacturing and installing lamp networks, constructing electricity transmission devices, so on
and so forth. To make a direct comparison to DACC, we only focus on the energy requirement
specifically for carbon capture. Therefore, we didn't include the energy costs associated with
engineering aspects, as the estimation of DACC's energy requirement does not include the
energy costs required for carbon transport, storage, and utilization. In this study, we also mainly
focus on the physical understanding of tropical forest ecosystem's responses to nighttime
artificial lighting, so we didn't have much discussion on engineering aspects (how such a
network of lamps could be constructed) as well as costs estimates. Nevertheless, the estimation
of additional energy costs and the engineering feasibility are important, and we will attempt to
address these issues in future studies.
As to the energy source, we assume this strategy only uses clean energy coming from solar, wind
or nuclear farms to avoid extra carbon emissions when providing light to forests.  In terms of
technical analysis, more clean energy can be acquired by deploying more low-carbon energy
generation plants across the globe (e.g., building large-scale solar and wind farms in the Sahara
Desert(Li et al., 2018)). In terms of economic analysis, however, both DACC and this strategy
are energetically and financially costly, and therefore, are unrealistic at present(Chatterjee and
Huang, 2020). Moreover, even if the clean energy generation capacity increases, we cannot
expect the global clean energy supply to only be invested to absorb $CO_2$. Nevertheless, if society
has urgency to intervene in Earth's climate by removing $CO_2$ from the atmosphere in the late half
of the 21th century, and/or an energy revolution realizes and we achieve the status of a
significant surplus of clean energy, CDR would still be a powerful and effective climate
mitigation strategy.
Another critical negative impact of this strategy is the potential threat to local wildlife and
biodiversity. Tropical forests are the repository of a large proportion of Earth's biodiversity, and
many of the organisms in the tropics are nocturnal or crepuscular, with organisms and
interactions occurring in darkness. An extension of photoperiods could disrupt the habit of
nocturnal creatures and have unexpectedly large impacts on ecosystem biodiversity. In addition,
the disruption, disturbance and habitat fragmentation resulting from installing lights throughout
tropical forests and throughout the forest canopies could exacerbate the negative impacts of this
strategy. Given the potentially inverse relationship between more light at night and ecosystem
health, policy makers may consider extending the photoperiod to an appropriate level to increase
carbon sequestration meanwhile protecting local biodiversity from disastrous impacts. The
tradeoff between nighttime carbon sequestration and biodiversity preservation should be
rigorously evaluated and weight in the decision making process.
Overall, lighting up tropical forests at night has led to significant increase in carbon uptakes,
decrease in atmospheric $CO_2$ concentration, and suppression of global warming as simulated by
Earth System Model. However, it has strong side effects after the termination of nighttime
lighting. In addition, local ecosystem changes could have negative impacts on local wildlife.
Practical issues include the large demand for clean energy and the difficulties for
implementation. From a positive standing it might be treated as an emergency climate solution if
the society relies heavily on carbon removal to adjust the Earth's climate in the future. Paris
Agreement set climate goals to limit global warming to well below 2 degree Celsius and
preferably to 1.5 degree Celsius compared to pre-industrial levels(Lawrence et al., 2018). To
accomplish the Paris Agreement's climate goals, different engineering levels (lighting powers,
areas, and periods) might be needed under various anthropogenic emission scenarios, with high-
emission scenarios possibly requiring high engineering levels. This study investigated the highest
engineering level (lighting up global tropical forests at night with the optimal power) under a
low-emission scenario (see Methods). Further research is needed to investigate the relationship
between engineering levels and emission scenarios in the context of global climate goals set out
by the Paris Agreement(Lawrence et al., 2018).
Current geoengineering studies mainly focus on the evaluation of climate goals that a potential
solution might or might not accomplish; however, the changes in Earth's climate after
terminating a geoengineering measure tend to be overlooked. This study suggests the importance
of post-geoengineering analysis in geoengineering studies.


## 4. Methods

The CESM2 is an open-source community coupled model consisting of atmosphere, ocean, land, sea-ice, land-ice, river, and wave models that exchange states and fluxes via a coupler(Danabasoglu et al., 2020). In this study, we used standard CESM2 configurations and enabled all modules including the Community Atmosphere Model version 6 (CAM6), the Parallel Ocean Program version 2 (POP2) with an ocean biogeochemistry component, the Community Land Model version 5 (CLM5) with a land biogeochemistry component, CICE version 5.1.2 (CICE5), the Community Ice Sheet Model Version 2.1 (CISM2.1), the Model for Scale Adaptive River Transport (MOSART), and the NOAA WaveWatch-III ocean surface wave prediction model version 3.14 (WW3). The CESM2 is part of the Couple Model Intercomparison Project Phase 6 (CMIP6) core simulations as well as about 20 Model Intercomparison Projects (MIPs) within CMIP6. Extensive evaluation suggests that the CESM2 simulations exhibit agreement with satellite era observations of the climate mean state, seasonal cycle, and interannual variability, which has identified CESM2 as among the most realistic climate models in the world(Danabasoglu et al., 2020).

4.1 Historical control simulations from 2001 to 2014

CESM2 has published its official historical simulation datasets from 1850-2014 on the Earth System Grid Federation (ESGF; https://esgf-node.llnl.gov/search/cmip6). This study analyzed the historical simulation datasets of two members from 2001 to 2014 produced by the CESM2 esm-hist-BPRP case.

4.2 Future experiment and control simulations from 2015 to 2050

The selection of 2015 as the start year of the lighting experiment follows CMIP6 future scenario simulation rules. The future experiment simulations and control simulations were both based on the Shared Socioeconomic Pathways (SSP) 126 scenario(Riahi et al., 2017), which is a low-emission (low fossil fuel combustion and deforestation) scenario. The Earth's climate state under SSP126 is close to the current climate state with respect to high-emission scenarios. Therefore, the selection of SSP126 controlled variables and allowed us to see how the lighting experiment along influences tropical forest carbon fluxes and climate. This study ran the CESM2 esm-SSP126-BPRP case with the official restart files from historical simulations (esm-hist-BPRP case). Thus, no model spin up was needed. All simulations were forced with specified greenhouse gas emissions rather than atmospheric greenhouse gas concentrations, so the atmospheric $CO_2$ concentration was prognostic and land and ocean carbon cycles feed back on

atmospheric $CO_2$. Each simulation has a nominal horizontal resolution of 1° and has two
members created from small perturbations to initial climate states to estimate uncertainties.
4.3 The lighting experiment design
The authors modified the radiation module (Rapid Radiative Transfer Model for General
circulation models, RRTMG) of CESM2 to add diffuse visible light to tropical forest canopy at
night. CESM2 determines if a grid column is at daytime or nighttime by calculating its cosine
(solar zenith angle) at each time step. A negative cosine indicates the grid column is at nighttime
and the incoming solar radiation would be assigned with zero. A positive cosine indicates
daytime, and the cosine value would be used to calculate incoming solar radiation. The land
module then calculates and passes the surface albedo to the atmosphere module and the
atmosphere module calculates the radiation fluxes with the surface albedo and the model-
calculated incoming solar radiation. We made modifications in all active modules to switch the
sign of tropical forests' cosine from negative to positive when tropical forests were at night. As a
result, all modules regarded tropical forests to be at daytime at every time step.
CESM2 divides the incoming solar radiation into four components: direct visible light, diffuse
visible light, direct near infrared light, and diffuse near infrared light. The authors assume that
the artificial light would be provided by a lamp network above the forest canopy and that trees
receive light from multiple directions. Therefore, the artificial light was specified as diffuse
visible light for simplification. In the model, we assigned the diffuse visible light component of
the incoming solar radiation with 100, 200, 300, or 400 and other components with 0. The
surface albedo was still calculated by the land module and passed to the atmosphere module. The
radiation fluxes were then calculated by the model-calculated surface albedo and manually-
specified solar insolation.
4.4 The calculation of the energy requirement for capturing one ton of carbon
$E = (Power \times Area \times Hours)/Carbon$                  (2)
where E is the energy requirement for capturing one ton of carbon per year; Power is 200W/$m^2$
(nighttime lighting power); Area is the tropical forest area $10.71 \times 10^{10} m^2$ (CESM2 output);
Hours is the amount of nighttime lighting hours per year: $365 \times 11$; Carbon is the net carbon
uptake per year (Fig. 2-f) simulated by CESM2. There is no assumed data in this calculation.

## Code and Data Availability

CESM2 is an open-source community climate model preserved at

https://doi.org/10.1029/2019MS001916. All data have been included in the manuscript.

## Author contribution

XG designed the study and performed the simulations. XG, SL, DW, YL, BH, and AJ
contributed to the data interpretation. XG drafted the original version of the manuscript. SL and
DW reviewed and edited the manuscript.

## Competing interests

Authors declare that they have no competing interests.

## Acknowledgments

We would like to acknowledge high-performance computing support from Cheyenne
(doi:10.5065/D6RX99HX) provided by NCAR's Computational and Information Systems
Laboratory, sponsored by the National Science Foundation. We would like to acknowledge the
constructive comments and suggestions from William Wieder.

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
