# Peer review of "Exploration of a novel geoengineering solution: lighting up tropical forests at"

_Earth System Dynamics, 2021_

## Author Comment (AC3)

24-hour Amazonian Tropical Forest Soil Moisture Changes under Control Simulation. Soil moisture here represents the mass of water in the 10cm soil surface.

[Figure]

**Global Tropical Forest Soil Moisture Responses**

---

## Author Response (AR1)

**Note: author responses are underlined. All the line numbers in author responses refer to the marked-up manuscript, rather than the original manuscript or the revised one.**

Public Comment from Richard Rosen:

There are many problems with this proposed article, but let's start with the biggest ones. Line 165 states that the authors estimated the amount of energy needed to produce the light that would lead to one ton of CO2 being removed from the atmosphere, yet they do not show their calculations. These calculations and all data that are assumed in these calculations must be provided. They should also show the total amount of energy needed to produce the light in their scenarios per year.

**Thank you for your comments!**

**We apologize for missing this part in the manuscript. We have incorporated the calculations into the manuscript (lines 417-422).**

Secondly, they do not even mention where all this energy is going to come from, and how such a network of lamps as noted on line 306 could be constructed. How much ecological damage would that cause? How much additional energy would it take to manufacture and install such a network of lamps to yield the 200w per m-squared intensity they site? Where would all this energy come from, renewable electricity? The authors must answer all these kinds of questions and more that I have not thought of yet to make their scenario even remotely plausible. This all must be addressed in this article. Off hand, the entire scheme seems crazy, and the potential negative impacts only seem to have been partially addressed.

The authors should also address the basis for their cost estimate per ton of CO2 removed, which is not given, if they have made a cost estimate.

**Other than the direct lighting energy, this strategy requires additional energy associated with manufacturing and installing lamp networks, constructing electricity transmission devices, so on and so forth. To make a direct comparison to DACC, we only focus on the energy requirement specifically for carbon capture. Therefore, we didn't include the energy costs associated with engineering aspects, as the estimation of DACC's energy requirement does not include the energy costs required for carbon transport, storage, and utilization. We assume this strategy only uses clean energy coming from solar, wind or nuclear farms. In this study, we also mainly focus on the physical understanding of tropical forest ecosystem's responses to nighttime artificial lighting, so we didn't have much discussion on engineering aspects (how such a network of lamps could be constructed) as well as costs estimates. Nevertheless, the estimation of additional energy costs and the engineering feasibility are important, and we will attempt to address these issues in future studies. We have added the discussion to the manuscript. (lines 306-316)**

**Another critical aspect is the potential negative impacts, or side effects of this strategy. We quantified the local temperature increase and the CO2 outgassing rate and amount after a sudden and sustained termination of the lighting experiment. However, the ecological damage (e.g., damage to local wildlife and biodiversity) is hard to quantify, although it is important. Therefore, our discussion is more qualitative on this aspect. We have made a supplement to this part in Discussion. (lines 329-340)**

Thanks, but from your figures it appears that the amount of generating capacity needed to illuminate all the square meters of tropical forest would be about 22,000 GW, or about 20 times the total generating capacity of the United States. If my calculation is correct, this is hardly a practical approach to removing CO2 from the air. That is why I asked you in my first comment how you propose to provide all this electricity to run the lights shining on all the needed tropical forest to remove as much CO2 as you propose removing by this scheme.

**Large clean energy requirements have always been a hurdle to large-scale deployment of any CDR (Carbon Dioxide Removal) techniques, including DACC and the strategy we discuss in this study. In terms of technical analysis, we can get more clean energy by deploying more low-carbon energy generation plants across the globe (e.g., building large-scale solar and wind farms in the Sahara Desert (Li, et al. 2018)). In terms of economic analysis, however, both DACC and this strategy are energetically and financially costly, and therefore, are unrealistic at present (Chatterjee, et al. 2020). Moreover, even if the clean energy generation capacity increases, we cannot expect the global clean energy supply to only be invested to absorb CO2. Nevertheless, if society has urgency to intervene in the Earth's climate by removing CO2 from the atmosphere in the late half of the 21th century, and/or an energy revolution realizes and we achieve the status of a significant surplus of clean energy, CDR would still be a powerful and effective climate mitigation strategy. (We have added it to the manuscript, lines 317-328)**

Reviewer Comment from Anonymous Referee #1:

This manuscript studied a new geoengineering method – lighting tropical forest to enhance terrestrial carbon sink, using CESM2. Although this is a new idea, and it is interesting to research the forest response with model simulation, this manuscript is missing some important analysis on: (1) the impact on ecosystem. We cannot ignore the whole ecosystem but only focusing on the carbon cycle. (2) energy needed for lighting the forest. The manuscript only shows a figure with couple sentences on this topic. There is detailed calculation discussed. (3) model uncertainty. Models are built and designed for current plant phenology and plant physiology. Can the model simulate plant and soil chemistry without night? (4) some of the results. Results are not fully analyzed and discussed (see general comments below). I suggested to reject this manuscript, but encourage resubmit after addressing those important issues.

**Thank you for your comments! (1) In this study, we discussed the impacts of nighttime artificial lighting on energy balance (incoming radiation, latent heat, and sensible heat in Fig.1), climate (local surface air temperature, local precipitation, and global surface air temperature in Fig.2), and carbon sequestration. However, we realized that we did not give enough discussion to the negative impacts of this strategy, especially on ecosystem biodiversity. We have made a supplement to this part in our revisions (lines 329-340). (2) We apologize for missing this part in the manuscript. We have incorporated the calculations into the manuscript (lines 417-422). (3) We agree that model simulations have large uncertainties due to a lack of understanding of forests' physiological responses to nighttime lighting as we discussed in Discussion (lines 237-248). Ideally, small-scale field experiments should be conducted to get a better knowledge of forests' physiological responses to nighttime lighting, after which we modify modern models. However, the**

**physiological responses of tropical trees to longer photoperiods overall have received little attention, and field experiments are lacking. Numerical simulation is currently the only available tool for us, and it provides us with one possible outcome scenario of the lighting experiment. We expect to see more field experiments to be conducted in the future to improve our understanding of the ecosystem responses of tropical forests to longer photoperiods.**

General Comments:

1. More detailed literature review on plant response to longer light explosion.

   **We have made a supplement to the current introduction and provided a more detailed literature review on this aspect (lines 66-81).**

2. Why do you assign a random 0-1 value of cosine for tropical forest during night? If you provide a constant diffuse light at night, shouldn't be the cosine number and surface albedo constant?

   **Normally, cosine (solar zenith angle) is used in two places in each module. First, the sign of the cosine is used by the model to determine if a grid column is at daytime or nighttime. A negative cosine indicates the grid column is at nighttime and the incoming solar radiation would be assigned with zero. A positive cosine indicates daytime, and the cosine value would be used to calculate incoming solar radiation. In our case, a tropical forest grid column is normally at nighttime. We change the sign of the cosine from negative to positive to turn on the calculation of atmospheric and land processes in this grid column. The new cosine value is now used to calculate incoming solar radiation by the model. However, we don't really need the model-calculated incoming solar radiation as we have to specify the value of each component in incoming solar radiation manually. This is why the sign of cosine (solar zenith angle) matters while its absolute value does not. We may have not stated this point clearly in the manuscript and we have made clarification in the revision. (lines 399-416)**

3. Figure 1: why is the night time NEP even higher than daytime control? If 200 W/m2 provides maximum NEP, why during daytime in the control, the maximum NEP is at the time of 13:00-15:00? It is better to have a local time axis there. This also applies to Fig. S2 and Fig. S3

   **The nighttime NEP is higher than daytime because nighttime surface radiation is solely diffuse visible light while daytime surface radiation is composed of direct NIR (~16%), diffuse NIR (~30%), direct visible light (~15%), and diffuse visible light (~39%). We have added this point to the manuscript (lines120-123).**

   **During daytime in the control simulation, the maximum NEP shows up around 9:00-11:00 am (Fig. 1-b). It is not likely to be due to clouds according to the diurnal pattern of the surface downward shortwave radiation (Fig.1-a). We examined the diurnal curve of the soil moisture (the red dash line in Fig. 1-b), and it seems to be due to soil moisture stress. Soil moisture was consumed quickly in the morning which led to water stress for plant growth in the afternoon. The soil moisture pattern also explains the biased**

**distribution of daytime surface air temperature (Fig.1-c), and slightly biased daytime latent heat (Fig.1-d), and daytime sensible heat (Fig.1-e). (lines 125-131)**

**We have changed x-axis in Fig. 1, Fig. S2, and Fig. S3 to local time. Thanks for the good suggestion.**

4. It is not clear how the diffuse light is added during day time. If they are not added during daytime, then why in Figures are there colored lines (indicating adding difference amount of diffuse radiation)?

   **No radiation was added to forests during the daytime. The nighttime radiation influences energy balance and atmospheric processes, and may have exerted impacts on cloud processes, which leads to slight differences in daytime surface radiation.**

5. Why does night NEP show different responses to added diffuse radiation in different regions (Figure 1, Fig. S2 and S3)?

   **Overall, Amazonian, African, and Asian tropical forests show similar nighttime NEP responses to nighttime radiation. Slight differences (e.g. blue and yellow lines in Fig. 1 and Fig. S2) may be due to the divergent ambient surface air temperatures (Amazonian tropical forests have an overall higher surface air temperature with respect to the other two tropical forests) or soil moisture conditions.**

6. If there are more burning materials after lighting, why wildfire simulated decreases? (Figure 2)

   **The wildfire risk estimation in CESM2 is associated with soil moisture. We examined the long-term soil moisture changes and found that nighttime lighting experiments increased soil moisture because of enhanced precipitation in tropical forests. The global tropical forest soil moisture changes have been added to Fig.2 (Fig.2-i). Therefore, increased soil moisture would reduce wildfire risks despite the increase of biomass and potential burning materials. We have made modifications to the manuscript.(lines 145-147)**

7. Why does GPP drop to lower than the control level after termination? (Figure 2, Line 182)

   **The annual gross primary production dropped quickly, ultimately reaching levels that were even lower than the control period due to a reduction in atmospheric CO2 (CO2 has a fertilization effect in the model). (lines 222-225)**

8. Figure 2: The shaded area in f is confusing. Why does the local temperature go back to the control level after lighting terminated, but the global averaged temperature keeps lower than the control? If this is CO2 effect, how could the local temperature is back to the level of control?

   **We have removed the shaded area in Fig.2-f to improve visualization. But the corresponding analysis is still in the manuscript. (lines 230-232)**

**As to the second question, we attribute it to two possible reasons. First, different regions tend to have diverse air temperature responses to global CO2 changes. Arctic regions show a much larger temperature increase in response to CO2 increase, while the temperature increase in tropical regions is not that significant. Similarly, the CO2 reduction may exert diverse impacts on temperature changes in different regions. Second, the temperature change in tropical forests at the termination of the experiment is controlled by two factors in this study, decreased incoming shortwave radiation and reduced CO2. The former has a much larger impact on the local energy balance than the latter. Therefore, the influence of CO2 reduction on local tropical forests is not as large as on the global scale. We have added the above discussion to the manuscript. (lines 284-295)**

9. Line 143: this is not correct. The energy consumption for lighting the forest is not caculated.

   **We understand the reviewer's concern which is mainly associated with the potential fossil fuel consumption when providing light to forests. If fossil fuel is used to provide energy for nighttime lighting, extra carbon would be emitted, and our conclusion could be wrong. In this study, we assume this strategy only uses clean energy coming from solar, wind, or nuclear farms. Therefore, no additional carbon emissions would be happening. As to where the clean energy would be coming from, we have added a detailed discussion to the manuscript. (lines 296-328)**

10. Line 169 (Figure 5): how does this calculate?

    **We have incorporated the calculations into the manuscript (lines 417-422).**

**Thank you very much for your constructive comments and suggestions again!**

Reviewer Comment from Jessica Gurevitch:

This is an interesting and creative approach to carbon capture, enhancing the natural process of photosynthesis by forest trees by extending lighting to nighttime. Unfortunately there are two major flaws with this approach. The Earth system is currently facing two major catastrophic changes: climate change and rapid, profound and extensive biodiversity loss. The first major problem with this proposed geoengineering solution to atmospheric carbon reduction is the effects that eliminating night in tropical forests would have on global biodiversity. Humans are intimately interconnected with the lives of other organisms, and threats to biodiversity have very large implications for human well-being as well. Over hundreds of millions of years, organisms, ecological communities, and ecosystems have evolved in response to the day/night regimes in different parts of our planet. Although humans are largely diurnal, and city-dwellers may be unaware of life in forests at night, there is a rich and central role of nighttime activities for many other living things. Tropical forests are the repository of a large proportion of Earth's biodiversity, and many of the organisms in the tropics are nocturnal or crepuscular, with organisms and interactions occurring in darkness. Bats, jaguars, moths, many fish, reptiles and amphibians, hunt, mate compete and otherwise interact at night. Bats pollinate tropical trees and lianas at night, resulting in the ability of these plants to reproduce. No reproduction, no trees, no forest.

Migrating birds often travel at night, using the stars for direction. Plants use daylength to initiate and regulate reproduction and other physiological functions. While the authors devote a single sentence to "impacts on local wildlife" this casual dismissal displays either disinterest or ignorance of the magnitude of threats to global biodiversity, particularly in the tropics. In addition to nixing night, the disruption, disturbance and habitat fragmentation that would result from installing lights throughout tropical forests and throughout the forest canopies would be unimaginably huge. This would greatly exacerbate the negative impacts of the night-removal proposed. I recommend a simple google search of "nocturnal animals in tropical rainforests" to learn more. One could spend a lifetime learning about tropical forests at night. The second major issue is, would this even work? As mentioned in the paper, depletion of local soil nitrogen (and phosphorus) and water could greatly curb the promise of turbo-charged photosynthesis. Even with increased rainfall, water deficits at small scales—the scales of the root systems of trees—may occur, particularly if rainfall occurs in intense storms, increasing runoff (another possible problem, which admittedly is also a problem with anthropogenic climate change). Are these lights going to work flawlessly for long periods of time, or are panels going to break, malfunction, be lost in storms and fires, be stolen, and be lost as deforestation continues throughout the tropics? Seems pretty certain that these things will occur and that they are not taken into account in these models. In short, this is a creative approach to "thinking outside the box" about reducing the enormity of the impacts of climate change, but too few things have been taken into account to fully understand the limitations to the predicted successes and even more important, the very dangerous potential consequences of eliminating night in tropical forests.

A specific comment: [Abstract] Plants do not "primarily" conduct photosynthesis during the daytime, they only conduct photosynthesis during daylight.

**Thank you for your comments! We apologize for not giving enough discussion to the potentially negative impacts of this strategy, especially on local wildlife and ecosystem biodiversity, and we realized that a simple sentence "local ecosystem changes could have negative impacts on local wildlife" in the manuscript must have left people an impression of the authors' disinterest or ignorance of the importance of biodiversity. Please believe us we never attempt to ignore the importance of global biodiversity. We do believe that biodiversity conservation is critical to the sustainable development of both natural and human systems.**

**When proposing the idea of lighting up tropical forests at night as a potential climate mitigation strategy, we don't mean to 100 percent eliminate night in tropical forests. We may consider extending the photoperiod to an appropriate level to increase carbon sequestration meanwhile protecting local biodiversity from disastrous impacts. Nevertheless, a longer photoperiod and shortened nighttime could also have unexpectedly large impacts on local wildlife and biodiversity. We have made a supplement to this part and added more discussion about the potential threats to local wildlife and ecosystem biodiversity in the revisions. (lines 329-340)**

**Thank you very much for your constructive comments and suggestions again!**